# Integrating Environmental Data with Medical Data in a Records-Linkage System to Explore Groundwater Nitrogen Levels and Child Health Outcomes

**DOI:** 10.3390/ijerph20065116

**Published:** 2023-03-14

**Authors:** Christine M. Prissel, Brandon R. Grossardt, Gregory S. Klinger, Jennifer L. St. Sauver, Walter A. Rocca

**Affiliations:** 1Division of Epidemiology, Department of Quantitative Health Sciences, Mayo Clinic, Rochester, MN 55905, USA; 2Division of Epidemiology and Community Health, School of Public Health, University of Minnesota, Minneapolis, MN 55455, USA; 3Division of Clinical Trials and Biostatistics, Department of Quantitative Health Sciences, Mayo Clinic, Rochester, MN 55905, USA; 4Water Resources Center, University of Minnesota Extension, Minneapolis, MN 55455, USA; 5The Robert D. and Patricia E. Kern Center for the Science of Health Care Delivery, Mayo Clinic, Rochester, MN 55905, USA; 6Women’s Health Research Center, Mayo Clinic, Rochester, MN 55905, USA; 7Department of Neurology, Mayo Clinic, Rochester, MN 55905, USA

**Keywords:** epidemiology, spatial methods, adolescent health, child health, groundwater nitrogen, records-linkage system

## Abstract

**Background**: The Rochester Epidemiology Project (REP) medical records-linkage system offers a unique opportunity to integrate medical and residency data with existing environmental data, to estimate individual-level exposures. Our primary aim was to provide an archetype of this integration. Our secondary aim was to explore the association between groundwater inorganic nitrogen concentration and adverse child and adolescent health outcomes. **Methods**: We conducted a nested case-control study in children, aged seven to eighteen, from six counties of southeastern Minnesota. Groundwater inorganic nitrogen concentration data were interpolated, to estimate exposure across our study region. Residency data were then overlaid, to estimate individual-level exposure for our entire study population (*n* = 29,270). Clinical classification software sets of diagnostic codes were used to determine the presence of 21 clinical conditions. Regression models were adjusted for age, sex, race, and rurality. **Results**: The analyses support further investigation of associations between nitrogen concentration and chronic obstructive pulmonary disease and bronchiectasis (OR: 2.38, CI: 1.64–3.46) among boys and girls, thyroid disorders (OR: 1.44, CI: 1.05–1.99) and suicide and intentional self-inflicted injury (OR: 1.37, CI: >1.00–1.87) among girls, and attention deficit conduct and disruptive behavior disorders (OR: 1.34, CI: 1.24–1.46) among boys. **Conclusions**: Investigators with environmental health research questions should leverage the well-enumerated population and residency data in the REP.

## 1. Introduction

Several previous environmental health studies have examined environmental exposures and health outcomes at an aggregated level [1,2,3]. Aggregation is often used because detailed residency data are unavailable through research databases in the United States [4]. Other studies have avoided the need to aggregate environmental data, by collecting individual exposure or residency data [5,6]. A common approach to collecting exposure and residency data, is through participant or household surveys [6,7]. With survey response rates often concerningly low, this method of data collection can be problematic [8]. Moreover, researchers employing survey methods must consider the generalizability of study results when notable proportions of the target population are not responding, and the potential recall bias from survey responders [9]. Other studies have employed samplers to ascertain location and individual exposures [10]. These studies are typically financially and time burdensome and therefore may not be practical for all environmental health research questions. When sufficient environmental data are available, spatial methods can be used to estimate individual exposure [11]. However, inconsistent or missing individual residency and health data can remain obstacles (e.g., unknown residency, inconsistent follow-up, inability to verify residency over time, unknown disease status, etc.). In part due to these obstacles, relatively few observational environmental health studies have used a well-enumerated population, with robust residency and health data, within the United States (US). The Rochester Epidemiology Project (REP) offers a unique infrastructure to overcome these obstacles and contribute to the field of environmental public health tracking (EPHT).

The REP is a comprehensive medical records-linkage system, that covers nearly all Olmsted County residents and most residents in the 26 surrounding counties [12,13,14]. The REP’s well-enumerated population and the processes of verifying and collecting residency data of a dynamic population over time, provides a rare opportunity to integrate environmental and medical data [14,15]. Few REP studies have examined the environment in which the REP population resides and its relationship to various health outcomes [16,17,18,19]. MacLaughlin and colleagues recently examined human papillomavirus (HPV) vaccine uptake disparities, by estimated individual-level socioeconomic status [19]. Although this study did not examine environmental exposures, it did leverage the residential data available for the REP population [19]. Other studies have examined similar research questions at a group-level [16,17,18]. Rutten et al. estimated the odds of HPV vaccine initiation and series completion, based on geographical location, after adjusting for both US Census block and group level covariates [16]. Another study used participant residency to assign a corresponding US Census block group area deprivation index (ADI) [17]. REP researchers have also examined the association between US Census block group ADI and the likelihood of HPV vaccine initiation and completion [18]. Although these studies primarily focused on neighborhood demographic data, they have paved the way for the REP to emerge into the field of environmental epidemiology [16,17,18,19]. Currently, no environmental data are stored within the REP infrastructure [12,20]. This manuscript provides an example of how environmental data and spatial methods can be leveraged to estimate the effect of individual-level environmental exposures on clinically relevant outcomes, in a well-enumerated population, with robust residency and health data. No study to date has used the REP infrastructure in conjunction with environmental data to estimate individual-level environmental exposures and to examine a research question of interest.

Nitrogen contamination of groundwater has become an increasing problem [21,22]. Inorganic nitrogen in the groundwater is both a direct health determinant and a proxy for other contaminants of concern [22]. These contaminants are often associated with human use of pesticides, a myriad of chemicals and chemical mixtures with the purpose of killing unwanted creatures, plants, and fungi [7,21,23]. High nitrogen contamination of groundwater is one mechanism by which environmental changes are reflective of human pesticide use [21]. Exposure to pesticides has been linked to numerous adverse health outcomes [7,24,25,26,27,28,29]. An increased risk of various cancers, diabetes, poor neurologic and cardiovascular health, endocrine disruption, and allergic effects have been linked to pesticide exposure [24,26,29,30,31]. Although children and adolescents are resilient, they are not impervious to the adverse health outcomes of pesticide exposure [7,32]. Previous studies have provided evidence of the harmful effects of pesticides on child and adolescent health [28,31,33,34,35,36,37,38,39]. However, studies of pesticide and fertilizer exposure among the general child and adolescent population have been sparse [25]. Furthermore, there have been limited studies examining the association of pesticide exposure and child and adolescent health outcomes within southeastern Minnesota—an area including both urban and rural regions [40]. 

This study explored the relationship in children and adolescents between groundwater inorganic nitrogen concentrations and 21 medical conditions, defined by the clinical classifications software (CCS) [41], to provide a proof-of-concept example of integrating address data and environmental data to estimate individual-level environmental exposures. This proof-of-concept, of how to efficiently estimate individual-level environmental exposure, will help advance EPHT and public health. The processes described in this study can be used to improve environmental hazard surveillance, help identify environmental health disparities, and inform targeted public health interventions for communities disproportionately affected by environmental hazards.

The objectives of this study are twofold. Our primary objective was to provide a proof-of-concept example, integrating REP data and environmental data to estimate individual-level environmental exposure. Our secondary objective was to explore the association between groundwater inorganic nitrogen concentration (a marker of pesticide use) and child and adolescent health outcomes, using two groundwater data sources and a well-defined REP cohort.

## 2. Materials and Methods

### 2.1. Study Population

We conducted a nested case-control study in southeastern Minnesota, using Minnesota Department of Agriculture (MDA), Olmsted County Public Health Services (OCPHS), and REP data, to explore the association between groundwater inorganic nitrogen concentration (mg/L), as a marker for fertilizer and pesticide exposure, and adverse health outcomes in children and adolescents, aged 7–18 years. A six-county area was identified in southeastern Minnesota (Olmsted, Goodhue, Filmore, Houston, Wabasha, and Winona counties), based on where MDA and OCPHS groundwater data overlapped with the REP region. MDA and OCPHS data were used to estimate inorganic nitrogen concentration (mg/L) exposure at the individual-level. 

We determined the complete resident population of all persons aged 7 to 18 years old on 1 January 2017, using the personal timelines available through the REP census [12,20]. Further exclusions to the population were made because parents or guardians (or adolescents themselves in the case of the 18-year-olds) had denied use of medical record data for research, under Minnesota research authorization [42]. Furthermore, persons were excluded when we were unable to link their mailing address with a geolocation (latitude and longitude), or when they had limited medical record information available in the 5-year period before 1 January 2017, thus limiting our ability to define case status for the case-control study design (see Figure 1) [13,15,20].

### 2.2. Inorganic Nitrogen Concentration

Inorganic nitrogen naturally occurs in groundwater [43]; however, nitrogen levels of ≥3 mg/L suggest human sources have contaminated the groundwater [23]. Groundwater inorganic nitrogen [nitrate (NO_3_) + nitrite (NO_2_)] concentrations were used as a marker of topsoil fertilizer and pesticide use. MDA groundwater samples from 2007–2018 (*n* ~ 516) and OCPHS groundwater samples from 2007–2017 (*n* ~ 5608) were used. Both MDA and OCPHS groundwater sample data contained inorganic nitrogen concentrations [nitrate (NO_3_) + nitrite (NO_2_)] and sample location.

MDA and OCPHS inorganic nitrogen groundwater sample locations, with corresponding nitrogen concentrations, were plotted onto our six-county region using the ArcGIS (Geographic Information Systems) Pro 2.4 software. An ordinary kriging interpolation was used to estimate inorganic nitrogen concentrations at unsampled locations within our six-county region [44,45]. Previous studies have used ordinary kriging to estimate groundwater nitrate concentrations [44,45]. REP geolocation information (latitude and longitude) of residential addresses for our study population were obtained. Residential latitude and longitude were plotted onto our six-county region. We overlaid the residential location map layer onto our interpolated inorganic nitrogen concentration map layer (i.e., kriging), to estimate inorganic nitrogen concentration for each study participant. This process is illustrated in Figure 2. The individual inorganic nitrogen estimates were exported to a Statistical Analysis System (SAS) compatible file. Using the SAS 9.4 software, participants were categorized into high (≥3 mg/L) and low (<3 mg/L) inorganic nitrogen exposure, and association analyses were performed for each of the 21 conditions of interest [23]. High (≥3 mg/L) levels of inorganic nitrogen suggest human sources have contaminated the groundwater [23]. 

### 2.3. Clinical Conditions in the Study Population

Twenty-one CCS conditions were identified as outcomes of interest, using a multi-step process. The CCS diagnosis categorization system groups all possible International Classification of Diseases (ICD-9 and ICD-10) diagnosis codes into 285 clinically meaningful categories, across all body systems [41]. To select conditions most relevant to our pediatric population, we first conducted a thorough literature review of pesticide exposure and child–adolescent health outcomes. The evidence from the scientific literature was matched to specific CCS disease categories. Second, two authors reviewed the full list of CCS categories in detail and identified any additional conditions that could plausibly be related to pesticide exposure. Third, we excluded any CCS set of diagnostic codes that included the word “other” (e.g., other skin conditions), as these diagnosis categories are extremely broad and often contain a myriad of diagnoses [41]. Finally, only CCS conditions that could affect both boys and girls were examined. For each person, all electronically available diagnostic codes in the 5-year period before 1 January 2017 were included, to define the case status (i.e., diagnoses from 1 January 2012 to 31 December 2016). Persons were considered “cases” if they had two codes separated by more than 30 days, from among the codes in a defined CCS category. Controls were all persons in the full population of children and adolescents.

### 2.4. Statistical Analyses 

Controls were defined as REP persons who were at risk of one or more CCS conditions at baseline in our study population (*n* = 29,270). This definition allows controls to estimate the exposure prevalence in the entire study population at baseline [46,47,48]. A logistic regression model was used to estimate the effect of high vs. low groundwater inorganic nitrogen on various child and adolescent health outcomes (CCS conditions). Analyses were carried out overall, and separately for boys and girls. Overall analyses were adjusted for age (continuous, centered at the midpoint of 12.5 years), sex, non-white race, and rurality. Rural and urban categories were determined using the rural–urban commuting area (RUCA) codes, available from the Economic Research Service, part of the US Department of Agriculture [40]. These potential confounders were identified through a review of scientific literature and scientific reasoning. We reasoned that sex, as a surrogate for cultural and behavioral gender norms, may impact nitrogen exposure and the likelihood of seeking medical attention for health conditions [49,50,51]. If the child or adolescent was not seen at a hospital or clinic, then they would not have received a diagnostic code for a CCS condition, and therefore would not be counted as a case in this study. We reasoned that age may impact nitrogen exposure, through hygiene, play, and other behaviors that change throughout childhood and adolescent years, and that age may also impact our adverse health outcomes of interest, through the number of health care visits [49,52,53]. We used non-white race to adjust for potential racial disparities and race as a social construct, reasoning that historical and current racial disparities may impact adolescent nitrogen exposure and access to health care [53,54,55]. Lastly, we adjusted analyses for rurality (rural vs. urban) for two reasons. First, previous studies have shown a difference in clinical care utilization by rurality [56,57,58]. Second, other studies have provided evidence of potential different rural and urban behavioral norms related to pesticide and fertilizer use and exposure [39,58,59]. Plotted points were enlarged and skewed in Figure 2, to ensure confidentiality of study participants. Additionally, sex was examined as a possible effect modifier. Our hypothesis for this was two-fold. First, we hypothesized that sex may be an effect modifier, through biological and hormonal differences in the older age stratum (i.e., children who have experienced puberty). We did not anticipate this hypothesis to hold for the younger age stratum. Secondly, we hypothesized that sex may act as a surrogate variable for gender-based cultural and behavioral norms among children and adolescents. Analyses performed separately by sex strata were adjusted for age (continuous), non-white race, and rurality.

## 3. Results

### 3.1. Study Demographics

Our study population included 29,270 children, ranging in age from 7 to 18 years, who resided in a 6-county REP region of southeastern Minnesota on 1 January 2017 (Table 1). Children and adolescents were split approximately evenly between girls (48.8%) and boys (51.2%), and the sex distribution was similar across rural versus urban strata. Age was reported, for descriptive purposes, as two 6-year periods. However, in analyses, age was entered in the models as a continuous variable, centered at the mid-point of 12.5 years. Children within the age range of 7 to 12 years made up 52.9% of our population, whereas adolescents aged 13 to 18 made up 47.1%. The proportion of our population who was of non-white race varied substantially across urban and rural strata. In the urban strata, 22.8% identified as non-white race, whereas only 9.2% identified as non-white race in the rural strata. This resulted in an overall study distribution of 78.1% of participants identifying as white and 21.9% identifying as non-white. The distribution of high and low nitrogen exposure was markedly different between urban and rural strata. We found that 76.2% of children and adolescents living in rural areas had high exposure to nitrogen, whereas only 26.7% of children living in urban areas had high exposure to nitrogen.

### 3.2. Results for Child and Adolescent Boys and Girls

After adjusting for sex, age, rurality, and non-white race, the odds of chronic obstructive pulmonary disease and bronchiectasis among children with high nitrogen exposure was 2.38 (CI: 1.64–3.46) times the odds among children with low nitrogen exposure (Table 2, All children column). A positive association between high nitrogen exposure and chronic obstructive pulmonary disease and bronchiectasis was also observed in the sex specific strata; the odds ratios were 2.63 (CI: 1.69–4.10) for girls and 2.15 (CI: 1.38–3.35) for boys (Table 2, sex specific columns).

### 3.3. Sex Specific Results

In addition, after adjusting for age, rurality, and non-white race, we observed that the following conditions had notably different odds among boys and girls. The odds of thyroid disorders among girls with high nitrogen exposure was 1.44 (CI: 1.05–1.99) times the odds with low nitrogen exposure (Table 2, girls only column). Moreover, among girls, after adjusting for age, rurality, and non-white race, we observed that the odds of suicide and intentional self-inflicted injury with high nitrogen exposure was 1.37 (CI: >1.00–1.87) times the odds with low nitrogen exposure (Table 2, girls only column). For child and adolescent boys, after adjusting for age, rurality, and non-white race, we observed the odds of attention deficit conduct and disruptive behavior disorders with high nitrogen exposure was 1.34 (CI: 1.24–1.46) times the odds with low nitrogen exposure (Table 2, boys only column).

## 4. Discussion

The primary objective of this study was to provide an archetype of how REP medical and residency data, along with existing environmental data, can be leveraged to test clinically relevant hypotheses. We recognize that these geospatial methods are commonly used in environmental health studies. However, the novelty of this study is using these methods within a well-enumerated population, with robust residency and medical data [12]. Leveraging existing environmental data, and applying appropriate geospatial methods, will allow researchers to estimate individual-level environmental exposures at a significantly lower financial and time cost, compared to using study specific sampling. In part due to the limited environmental data maintained within the REP infrastructure, few studies to date have leveraged the REP robust medical and residency data. However, a few recent REP studies have investigated neighborhood demographics and associated health outcomes [16,17,18,19]. To advance public health, it is critical that clinically relevant studies begin to examine the impact of the places people work, live, and play [60]. In response, the REP has begun to develop the REP environmental exposure data dictionary. This searchable tool allows REP investigators to identify environmental data collected within the REP region and provides summary information about these environmental data (e.g., variables available, dates data were collected, data request form and contact, etc.).

Our exploratory results support further investigation of the association between high inorganic nitrogen concentrations and chronic obstructive pulmonary disease and bronchiectasis among children and adolescents. Specifically, for girls, further investigation is needed to understand the association of high nitrogen concentration, thyroid disorders, and suicide and intentional self-inflicted injury. For boys, we found evidence that the association of nitrogen concentration and attention deficit conduct and disruptive behavior disorders should be explored further.

A few CCS conditions showed reduced odds for those with high groundwater inorganic nitrogen concentrations (i.e., headache, including migraine, vision defects, dizziness or vertigo, nausea and vomiting, mood and developmental disorders). There is little to no biologic plausibility that high nitrogen concentrations are causing the reduced odds of these conditions [31,32,59]. Rather, we hypothesize that these odds reductions are due to decreased screening and diagnosing of these conditions in rural compared to urban regions. Healthcare behaviors and access differences among rural and urban residents impact screening, diagnosis, and treatment of these conditions. For example, if a child or adolescent was not diagnosed with nausea and vomiting at a healthcare facility, then they were not defined as a case, even if they had experienced nausea and vomiting. Moreover, a priori testing suggested that high nitrogen concentrations were highly correlated with rural residency.

A limitation of this study is that groundwater samples were not collected for the purpose of this study. We worked extensively with partners from the OCPHS environmental health department and MDA to understand the data and their collection processes. Investigators of future studies, that use these or other environmental data, are strongly encouraged to partner with owners of environmental data, to ensure accurate understanding, use, and interpretation. Groundwater nitrogen values may vary, due to geology differences throughout our study region [61]. This limitation was a motivator for analyzing nitrogen exposure as a dichotomous variable, based on levels that suggest that human sources have contaminated the groundwater (i.e., high, ≥3 mg/L and low, <3 mg/L) [23], rather than examining nitrogen exposure as a continuous measurement. Therefore, we were unable to assess a dose–response relationship for nitrogen levels and outcomes. Moreover, it is important to note that the groundwater samples used in this study are not synonymous with drinking well water samples. Other studies have used well data to examine nitrogen exposure from drinking water. However, our study focused on the health risks of indirect exposure, and therefore was not restricted to well water samples. Our exploratory results do not provide causal evidence for the association of high inorganic nitrogen concentrations and adverse child and adolescent health conditions. Some CCS conditions require pesticide breakdown products to cross the blood brain barrier [62,63]. High inorganic nitrogen levels suggest that human-made chemicals were used [23], but we did not examine levels of specific breakdown products (e.g., atrazine). However, our exploratory results can be used to identify signals for further investigation. Additionally, we did not adjust for education or family income, because education attainment correlates with the age of our study population, and family income was hypothesized to affect child exposure through residency (Figure A1 in Appendix A). Age and residency were adjusted for in our models. Finally, we excluded 8% of children because they did not have sufficient medical record information to assess medical conditions. A lack of medical record information may indicate recent movement into our population (immigration), extremely low to no utilization of, or access to, care, or children who are particularly healthy.

## 5. Conclusions

The primary purpose of this study was to provide an archetype of how REP medical and residency data, along with existing environmental data, can be leveraged to improve our understanding of the complex relationships between our environment and health. This knowledge may inform evidence-based public health policies and interventions, aimed at reducing the impact of environmental hazards on public health. The REP provides a unique infrastructure, with robust data on a well-enumerated population over time [12]. This longstanding infrastructure, combined with environmental data sources, allows for investigators to estimate individual-level environmental exposures. This study provides a path forward for future investigators to delve into the field of EPHT and examine additional environmental exposures and health outcomes. The REP has supported the effort to consolidate environmental data sources that overlap with the REP 27-county region for future investigator use. We anticipate that the proof-of-concept demonstrated in this study will be utilized in future research, to monitor environmental hazards, assess their impact on public health, and investigate their potential burden on the healthcare system.

## Figures and Tables

**Figure 1 ijerph-20-05116-f001:**
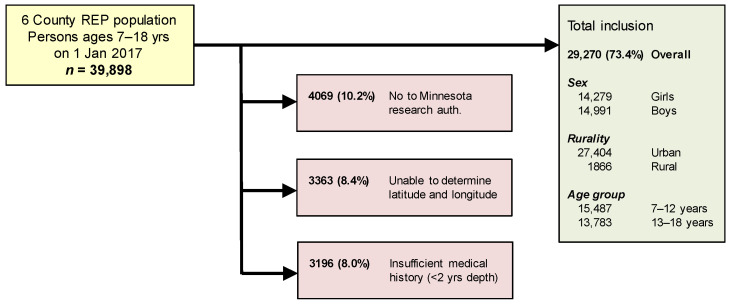
Inclusion of children and adolescents in study.

**Figure 2 ijerph-20-05116-f002:**
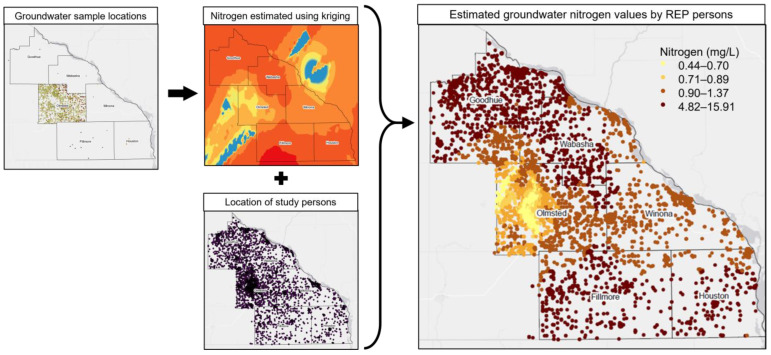
The combining of REP, MDA, and OCPHS data. Visual display of the generated map layers used to estimate person-level inorganic nitrogen concentrations. (**Left**) MDA and OCPHS groundwater nitrogen samples were mapped onto our six-county region. (**Middle top**) An ordinary kriging interpolation was used to estimate inorganic nitrogen concentrations within our six-county region. (**Middle bottom**) Study participants’ location of residence was mapped onto the six-county region. (**Right**) The kriging results and location of residence map layers were amalgamated to estimate inorganic nitrogen values for each study participant. Plotted points were enlarged and slightly jittered, to ensure confidentiality of study participants.

**Table 1 ijerph-20-05116-t001:** Descriptive characteristics of children and adolescents included in study.

	Urban *		Rural *		All Persons	
Characteristic	*n*	(%) †	*n*	(%) †	*n*	(%) †
**Full cohort of all persons †**	27,404	93.6	1866	6.4	29,270	100
**Sex**						
Female (girls)	13,369	48.8	910	48.8	14,279	48.8
Male (boys)	14,035	51.2	956	51.2	14,991	51.2
**Age group ‡**						
7 to 12 years	14,525	53.0	962	51.6	15,487	52.9
13 to 18 years	12,879	47.0	904	48.4	13,783	47.1
**Race**						
White	21,154	77.2	1695	90.8	22,849	78.1
Non-White	6250	22.8	171	9.2	6421	21.9
**Nitrate exposure (risk factor)**						
Low exposure (<3 mg/L)	20,086	73.3	445	23.8	20,531	70.1
High exposure (≥3 mg/L)	7318	26.7	1421	76.2	8739	29.9
**Medical conditions (case status)**						
Thyroid Disorders	233	0.9	11	0.6	244	0.8
Epilepsy, Convulsions	296	1.1	28	1.5	324	1.1
Headache, Including Migraine	1154	4.2	72	3.9	1226	4.2
Blindness and Vision Defects	3231	11.8	124	6.6	3355	11.5
Inflammation, Infection of Eye (Except that Caused by TB or STD)	1044	3.8	48	2.6	1092	3.7
Conditions Associated with Dizziness or Vertigo	136	0.5	9	0.5	145	0.5
Pneumonia (Except that Caused by TB or STD)	222	0.8	7	0.4	229	0.8
Acute Bronchitis	141	0.5	8	0.4	149	0.5
Chronic Obstructive Pulmonary Disease and Bronchiectasis	125	0.5	10	0.5	135	0.5
Asthma	2256	8.2	141	7.6	2397	8.2
Skin and Subcutaneous Tissue Infections	577	2.1	40	2.1	617	2.1
Syncope	107	0.4	17	0.9	124	0.4
Nausea and Vomiting	642	2.3	30	1.6	672	2.3
Malaise and Fatigue	335	1.2	19	1.0	354	1.2
Allergic Reactions	1692	6.2	83	4.4	1775	6.1
Anxiety Disorders	1866	6.8	102	5.5	1968	6.7
Attention Deficit Conduct and Disruptive Behavior Disorders	3221	11.8	186	10.0	3407	11.6
Developmental Disorders	756	2.8	42	2.3	798	2.7
Mood Disorders	1449	5.3	77	4.1	1526	5.2
Substance-Related Disorders	191	0.7	11	0.6	202	0.7
Suicide and Intentional Self-Inflicted Injury	247	0.9	19	1.0	266	0.9

* Urban and rural categories were determined using the Rural-Urban Commuting Area (RUCA) codes available from the Economic Research Service, part of the US Department of Agriculture (https://ers.usda.gov/). Percentages for urban and rural overall are across rows. † Percentages are reported within columns except for the first row, where they are reported across the row. ‡ Age was reported for descriptive purposes as two 6-year periods. However, in analyses, age was entered in the models as a continuous variable centered at the mid-point of 12.5 years.

**Table 2 ijerph-20-05116-t002:** Association analyses of 21 conditions with inorganic nitrogen levels, overall and separately in sex strata.

		Exposure (Risk Factor)		All Children		Boys Only		Girls Only		Sex
		Low Nitrogen	High Nitrogen	Odds Ratio †		Odds Ratio †		Odds Ratio †		Interaction
Group *	Total *n*	*n* (%)	*n* (%)	OR (95% CI) ‡	*p*-Value	OR (95% CI) §	*p*-Value	OR (95% CI) §	*p*-Value	*p*-Value ‖
** *Full population of children* **	29,270	20,531 (70.1)	8739 (29.9)	1 (reference)	---	1 (reference)	---	1 (reference)	---	---
** *Case status* **										
Thyroid Disorders	244	168 (68.9)	76 (31.1)	1.06 (0.80–1.40)	0.69	0.69 (0.45–1.05)	0.08	1.44 (1.05–1.99)	0.02	0.002
Epilepsy, Convulsions	324	239 (73.8)	85 (26.2)	0.84 (0.64–1.11)	0.23	0.76 (0.53–1.09)	0.13	0.93 (0.66–1.31)	0.69	0.33
Headache, Including Migraine	1226	896 (73.1)	330 (26.9)	0.83 (0.73–0.94)	0.004	0.66 (0.55–0.79)	<0.001	1.00 (0.86–1.17)	0.97	<0.001
Blindness and Vision Defects	3355	2527 (75.3)	828 (24.7)	0.86 (0.80–0.93)	<0.001	0.82 (0.74–0.91)	<0.001	0.90 (0.81–0.99)	0.04	0.17
Inflammation, Infection of Eye (Except that Caused by TB or STD)	1092	797 (73.0)	295 (27.0)	0.92 (0.80–1.05)	0.22	0.97 (0.82–1.15)	0.71	0.87 (0.72–1.04)	0.13	0.34
Conditions Associated with Dizziness or Vertigo	145	112 (77.2)	33 (22.8)	0.64 (0.42–0.97)	0.04	0.58 (0.33–0.99)	0.046	0.71 (0.42–1.20)	0.20	0.55
Pneumonia (Except that Caused by TB or STD)	229	177 (77.3)	52 (22.7)	0.79 (0.57–1.08)	0.14	0.79 (0.52–1.18)	0.25	0.78 (0.51–1.20)	0.26	0.99
Acute Bronchitis	149	98 (65.8)	51 (34.2)	1.24 (0.88–1.76)	0.22	1.43 (0.93–2.18)	0.10	1.05 (0.66–1.67)	0.85	0.28
Chronic Obstructive Pulmonary Disease and Bronchiectasis	135	68 (50.4)	67 (49.6)	2.38 (1.64–3.46)	<0.001	2.15 (1.38–3.35)	0.001	2.63 (1.69–4.10)	<0.001	0.41
Asthma	2397	1754 (73.2)	643 (26.8)	0.89 (0.82–0.98)	0.01	0.97 (0.87–1.08)	0.57	0.81 (0.72–0.92)	0.001	0.02
Skin and Subcutaneous Tissue Infections	617	416 (67.4)	201 (32.6)	1.06 (0.89–1.25)	0.54	1.21 (0.99–1.50)	0.07	0.89 (0.70–1.12)	0.32	0.03
Syncope	124	76 (61.3)	48 (38.7)	1.23 (0.83–1.82)	0.31	1.01 (0.60–1.69)	0.97	1.45 (0.92–2.30)	0.11	0.21
Nausea and Vomiting	672	526 (78.3)	146 (21.7)	0.75 (0.62–0.90)	0.002	0.67 (0.52–0.86)	0.002	0.83 (0.65–1.06)	0.13	0.18
Malaise and Fatigue	354	256 (72.3)	98 (27.7)	0.85 (0.67–1.08)	0.19	0.50 (0.34–0.74)	0.001	1.21 (0.93–1.59)	0.16	<0.001
Allergic Reactions	1775	1320 (74.4)	455 (25.6)	0.89 (0.80–0.99)	0.03	0.89 (0.77–1.02)	0.09	0.89 (0.77–1.02)	0.10	0.99
Anxiety Disorders	1968	1493 (75.9)	475 (24.1)	0.69 (0.62–0.76)	<0.001	0.56 (0.48–0.65)	<0.001	0.81 (0.72–0.92)	0.001	<0.001
Attention Deficit Conduct and Disruptive Behavior Disorders	3407	2369 (69.5)	1038 (30.5)	1.02 (0.95–1.10)	0.58	1.34 (1.24–1.46)	<0.001	0.67 (0.60–0.75)	<0.001	<0.001
Developmental Disorders	798	614 (76.9)	184 (23.1)	0.72 (0.60–0.86)	<0.001	0.90 (0.73–1.10)	0.30	0.53 (0.41–0.68)	<0.001	0.001
Mood Disorders	1526	1148 (75.2)	378 (24.8)	0.74 (0.66–0.83)	<0.001	0.48 (0.39–0.57)	<0.001	1.02 (0.89–1.16)	0.81	<0.001
Substance-Related Disorders	202	160 (79.2)	42 (20.8)	0.60 (0.42–0.85)	0.005	0.56 (0.35–0.91)	0.02	0.63 (0.40–0.99)	0.046	0.71
Suicide and Intentional Self-Inflicted Injury	266	188 (70.7)	78 (29.3)	0.93 (0.70–1.23)	0.60	0.49 (0.31–0.77)	0.002	1.37 (1.00–1.87)	0.047	<0.001

* All conditions were analyzed compared to the full population of all children. † Odds ratios were adjusted using Generalized Estimating Equation (GEE) techniques to account for the occurrence of children and adolescents present in both the control and case groups. The odds ratios are interpreted as the “risk of having high nitrogen in cases divided by the risk of having high nitrogen in the full group of children and adolescents.” A OR > 1.0 indicates that high nitrogen was more common in cases than in the reference group of all children. A OR < 1.0 indicates that high nitrogen was less common in cases than in the reference group of all children and adolescents. ‡ Overall analyses in boys and girls pooled were adjusted for sex (male = 1, female = 0), age (continuous, centered at 12.5 years), non-White race (non-white = 1, white = 0), and rurality (rural = 1, urban = 0). § Analyses separate in strata for boys and girls were adjusted for age (continuous, centered at 12.5 years), non-White race (non-white = 1, white = 0), and rurality (rural = 1, urban = 0). ‖ The *p*-value from the formal statistical test of whether results differed between boys and girls (i.e., sex interaction).

## Data Availability

Study data are not publicly available because of the sensitivity of distributing residency data. The sharing of de-identified data will be considered by application to the REP co-Principal Investigators on a case-by-case basis and protocol approval by an Institutional Review Board.

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
