# Peer review of "Integrating Environmental Data with Medical Data in a Records-Linkage System to Explore Groundwater Nitrogen Levels and Child Health Outcomes"

_ijerph, 2023, doi:10.3390/ijerph20065116_

Round 1
Reviewer 1 Report
The authors integrate environmental with medical data to study the association between groundwater inorganic nitrogen concentration and adverse health outcomes. Although their approach can provide an interesting contribution to the field, a more in depth statistical analysis has to be conducted to support the results described in the paper. See my comments below.
Line 181: “potential confounders were identified through review of scientific literature …”. Please also conduct a statistical analysis to check if these factors are confounding factors in your study. Furthermore, in case you have also measured other variables than those identified as confounders in the literature, please check if they are confounding factors by performing statistical tests. Compare logistic models including different factors with a model selection criterion (e.g. AIC). In this way you avoid overfitting or underfitting of your model.
Line 203: “ Analyses performed separately by sex strata were adjusted for age (continuous), non-White race, and rurality.” Same remark as above, confounders were not determined based on statistical analysis.
Line 237: “We observed that the odds of suicide and intentional self-inflicted injury with high nitrogen exposure was 1.37 (CI: 1.00-1.87) times the odds with low nitrogen exposure”. Based on the reported confidence interval, this is not significant (1 is in the CI) although we get a p-value lower than 0.05. I would therefore make a remark in the discussion that one has to be careful when interpreting the result.
Author Response
We thank Reviewer 1 for the detailed suggestions to improve our manuscript. Please see the attachment for point-by-point response to Reviewer 1's comments.

Reviewer 2 Report
A well written, thoughtful paper on the benefits of combining environmental and health data making the use of existing databases, registers and datasets to explore associations. The paper presents a case study to show the benefits of integrating medical and residency data and existing environmental data in an environmental epidemiology context. The paper gives a nice example of using ground water (GW) nitrogen concentrations and child health outcomes. The paper suggests the areas of investigations where this approach could be used, the limits to this approach and suggestions for future applications. This paper would be of great interest to the journal audience, showing the application of environmental public health and use and reuse of data and integration to help answer a public health question. This application of Environmental Public Health Tracking (EPHT) should be commended. For this reason I would recommend publication of the paper, subject to very minor amendments.
Suggested changes:
As mentioned above this paper is a classic example in the field of EPHT. No mention of EPHT is made in the paper, or keywords, so I would suggest adding this point in the introduction, and /or conclusion.
The title of the paper makes no reference to the case study of GW nitrogen and child health outcomes, so readers may not pick up this paper based on title searches alone. The title may want to include these keywords?
Abstract: Could ages of the children ( or groups) be referenced as opposed to just ‘boys’ and ‘girls’?
Introduction: Could you add a little more context to this population on water supplies? In general what proportion of the population utilise public and private water supplies? What is the size of the population potentially affected by inorganic nitrogen in public/private supplies? Can you describe sources more?
Line 97: How were these 21 medical conditions chosen or targeted?
Fig 1: If 8% were excluded due to insufficient medical records, does this include the ‘well people’ with no illness? Is this a limitation of the study design?
Fig 2: The maps on the left are very small and hard to read, and unlabelled.
Page 5- section 2.4. Was deprivation or social economic status data not available? Is this a limitation?
Table 1. Add a note about the calculation of percentages being proportions across the exposure categories (rows). For the first row (rurality split) a % for urban/rural could also be shown. The font of this table is very small.
Table 2: needs a little more explanation. What is the last column p value? By comparing odds in cases compared to odds in the whole population would dilute any measure of effect. Why was odds in cases not compared to odds in controls?
Section 3.3. Why only these outcomes are highlighted when there are others also of interest?
Limitations should include the fact that only 2 exposure groups were used, that does not allow any dose response relationship to be explored.
The paper is very well written and gives good arguments for the choice of methodology and application and findings so requires little change. Commendation to the authors for a great draft.
Author Response
We thank Reviewer 2 for the detailed suggestions to improve our manuscript and high marks. Please see the attachment for point-by-point response to Reviewer 2's comments.

Round 2
Reviewer 1 Report
The authors have sufficiently addressed my comments.